# New Generation of SF6-Free Medium-Voltage Switchgear for the Electrical Network: Stability and Toxicity Studies of Trans-1,1,1,4,4,4-Hexafluorobut-2-ene with $N_2$ Gas Mixture

**Maria Luz Alonso** [1,*] , **Ane Espinazo** [2] , **Rosa Maria Alonso** [1] , **Jose Ignacio Lombraña** [2] , **Jesús Izcara** [3] **and Josu Izaguirre** [3]

1  Analytical Chemistry Department, Faculty of Science and Technology, University of the Basque Country (UPV/EHU), Barrio Sarriena s/n, 48940 Leioa, Spain
2  Chemical Engineering Department, Faculty of Science and Technology, University of the Basque Country (UPV/EHU), Barrio Sarriena s/n, 48940 Leioa, Spain
3  Ormazabal Corporate Technology, Parque Empresarial Boroa, Parcela 24, 48340 Amorebieta-Etxano, Spain
*  Correspondence: marialuz.alonso@ehu.eus

**Abstract:** Binary gas mixture of $N_2$ and trans-1,1,1,4,4,4-hexafluorobut-2-ene (HFO4E) is presented as an alternative to SF6 in medium-voltage electrical equipment. Its stability was tested under different conditions. No change was observed in the gas mixture after a permanent AC voltage of 30 kV applied for two years or during the making operations with a standard load-break switch. The same behavior was obtained under dielectric tests, electrical arcs and temperature rise tests according to the IEC 62271-1:2011 standard. For all of these conditions, the concentration of HFO4E remains practically unchanged; there is no impact on the insulation properties of the system and the degradation products formed would not affect the health and the environment if there were leaks. In these studies, gas mixtures samples were analyzed by a validated methodology based on gas chromatography coupled to mass spectrometry and thermal conductivity detectors. Finally, an OECD TG 403 acute inhalation toxicity test was also carried out with the gas mixture aged after the mentioned making operations. None of the mice used in the toxicity test were affected after 4 h of exposition to an ambient air with 30,000 ppmv of the gas mixture.

**Keywords:** hydrofluoroolefins; SF6; stability; degradation products; LC50

## 1. Introduction

The European Commission is studying a new regulation for the use of SF6 in medium-voltage switchgear due to the environmental problems it presents; SF6 has a Global Warming Potential (GWP) of 25,200 [1,2]. In this sense, for switchgear the amended Directive of European Union 2019/1937 and repealing Regulation (EU) No 517/2014 establish January 2026 as the date for the prohibition of the use of fluorinated greenhouse gases with a GWP of 10 or more, or with a GWP of 2000 or more, unless evidence is provided that there is no suitable alternative [3]. In this regard, switchgear manufacturers have investigated over the last years the use of alternative gases to SF6 as natural origin gases ($CO_2$, $O_2$, $N_2$) or mixtures of these natural origin gases with new synthetic gases, known to offer better dielectric strength. In this sense, hydrofluoroolefins (HFOs) play an important role [4,5].

Fluorocarbons are, in general, less toxic than the corresponding chlorinated or brominated hydrocarbons. This lower toxicity may be due to greater stability of the C-F bond. Fluorocarbons have low level of toxicity, therefore, it has been possible to select fluorocarbons that are safe for the uses to which they are destined [6].

HFOs are unsaturated organic compounds used as refrigerants at home and in automotive air conditioning (A/C) systems. HFO refrigerants are categorized as having low GWP and therefore offer a more environmentally friendly alternative to chlorofluorocarbons

(CFCs), hydrochlorofluorocarbons (HCFCs) and hydrofluorocarbons (HFCs). HFOs are being developed as "fourth generation" refrigerants with the 0.1% GWP of HFCs. HFOs currently in use include 2,3,3,3-tetrafluoropropene (HFO-1234yf) and 1,3,3,3-tetrafluoropropene (HFO-1234zeE or HFO3E). 1-chloro-3,3,3-trifluoropropene (HFO-1233zd) is also under development [7].

In a previous work, HFO3E, alone and in mixtures with natural origin gases and with the fluoroketone perfluoro-(3-methylbutan-2-one), $CF_3C(O)CF(CF_3)_2$, was studied [8]. The results of this work would manage to be a good alternative to SF6 since these gas mixtures would be environmentally and economically more feasible for their use in medium-voltage switchgear (from 12 to 40.5 kV). Recently, hydrofluoroolefin trans-1,1,1,4,4,4-hexafluorobut-2-ene (HFO-1336mzzE or HFO4E) has been developed to be used as a foam expansion agent or a heat transfer fluid. It could also be used as an alternative insulation medium in medium-voltage switchgear in mixtures with natural origin gases offering advantages against HFO3E due to its nonflammability and dielectric strength (see Table 1).

**Table 1.** Physicochemical properties of HFO4E and HFO3E in comparison with SF6.

| Physicochemical Properties | SF6 [9] | HFO3E [9–11] Trans-1,3,3,3-Tetrafluoropropene R-1234ze(E) | HFO4E [9,12–14] Trans-1,1,1,4,4,4-Hexafluorobut-2-ene R-1336mzz(E) |
|---|---|---|---|
| Molecular structure |  |  |  |
| Molecular weight (g/mol) | 146 | 114 | 164.05 |
| Boiling Point (°C) | −63 | −19.4 | 7.5 |
| Dielectric strength (%SF6) | 100 | 85 | 159 |
| GWP (100 years) [1] | 25,200 | 1.37 | 17.9 |
| AcuteToxicity (LC50 4 h/rat, ppmv) | >500,000 | >200,000 | >25,400 |
| Chronic Toxicity (TWA, ppmv) | 1000 | 800 | 400 |
| Flammability | Nonflammable | Midly-flammable | Nonflammable |

In order to assess HFO4E gas mixtures as an appropriate alternative to SF6 in medium-voltage switchgear, it is necessary to study their stability in real field conditions of a medium-voltage switchgear (high electric field, electric arcs in the making operations of the load-break switch and temperature rise test). Therefore, the aim of this work is to study the chemical stability of HFO4E in combination with $N_2$ in switchgear at different operation conditions. One of them was performed at medium-voltage switchgear remaining with permanent 30 kV AC voltage in the Ormazabal Corporate Technology experimentation unit (UDEX) for two years. Another condition assayed consisted in the making operations at nominal current (630 A) and at short-circuit current (20 kA) carried out with a standard load-break switch in a medium-voltage switchgear. Gas mixtures were also kept under dielectric tests at 24 kV level (power-frequency withstand voltage of 50 kV and lightning impulse withstand voltage of 125 kV). Another assay was an analysis of the gas mixture after an electrical arc of 140 kJ caused by an electrical failure. Finally, the effect of a temperature rise test according to IEC 62271-200 was also performed in a prototype switchgear. An acute inhalation toxicity test (OECD TG 403) was also carried out with the gas mixture aged after the mentioned making operations.

The study of the behavior of the HFO mixtures and their possible decomposition products when medium-voltage electrical switchgear is used requires the development of a suitable analytical method for their monitoring. A simple and rapid one-dimensional chromatographic method with an optimal chromatographic resolution has been developed and validated for the quantitative analysis of gas mixtures and their possible decomposition

product characterization. Gas chromatography coupled to mass spectrometry (MS) and thermal conductivity (TCD) detectors [3,8,15,16] has been the analytical technique chosen for this purpose, based on the GC/MS-TCD method previously developed and validated in our research group [8].

## 2. Materials and Methods

### 2.1. Reagents

HFO4E was obtained from Chemours (Wilmington, NC, USA) with a purity $\geq$99.5%. Premier $N_2$ was supplied by Carburos Metalicos (Madrid, Spain) with a purity $\geq$99.992%. The decomposition products $C_2F_4$ and $C_3F_6$ at 500 ppmv were acquired from Air Liquid (Madrid, Spain) to be used as a gas reference standard. They are the only standards available on the market, of all the compounds identified in this work. Degradation products are crystal quality, recommended for calibrating analyzers and adjustment equipment for instrumentation, and with a manufacturing tolerance between 5% and 10% and an uncertainty between 2% and 5%.

### 2.2. Instrumentation

Gas chromatographic (GC) analysis was performed using a gas chromatograph Agilent 7820A (Santa Clara, CA, USA), equipped with mass spectrometry (MS) and thermal conductivity (TCD) detectors (5975C and G4332A, respectively).

### 2.3. Chromatographic Conditions

Gas chromatography equipment with two injectors was used in this work. One is connected to a column Poraplot Q (25 m $\times$ 0.25 mm $\times$ 8 µm) and a MS detector. The other injector is joined to two columns in serial connection, a capilar column HP-PlotQ (30 m $\times$ 0.53 mm $\times$ 40 µm) and the other one is a HP-Molesieve (30 m $\times$ 0.32 mm $\times$ 12 µm). These two columns are connected by a system of one valve that open and close to allow the gas sample to pass from one to the other, intermittently, until finally communicating with the TCD detector.

Gas samples were extracted from the cubicles of the switchgear and taken in 500 cm$^3$ tedlar bags from Vertex (Barcelona, Spain) to be analyzed at the same day by GC/MS and GC/TCD using a gas-tight Hamilton syringe glue-free HDHT (GR, Switzerland). Volumes of the inserted sample were 0.1 mL at 1:10 split ratio for TCD and 1 mL at 1:55 split ratio for MS, using a liner 5183–4647 (Agilent, CA, USA). Temperature of MS and TCD injectors was set at 200 °C and pressure at 2.4 psi and 14 psi, respectively. Oven temperature was programmed with an initial temperature of 40 °C for 5 min, followed by an increase at a rate of 10 °C·min$^{-1}$ to 75 °C. Finally, oven temperature was increased at a rate of 5 °C·min$^{-1}$ up to 200 °C.

The analysis of the mass spectra obtained allows identifying the decomposition products by chemical interpretation or comparison with the NIST14 mass spectra database. Mass spectra were collected from $m/z$ 15 to 300 (with an electronic impact ionization source of 70 eV).

### 2.4. Characterization of HFO4E Standard

Firstly, the chromatographic behavior of HFO4E was studied and the existence of impurities was tested. For this purpose, the gas standards were injected in the chromatographic system in the conditions previously reported for the analysis of mixtures of dielectric gases by GC/MS-TCD [8]. The analytical method was validated for the quantitative analysis of HFO4E following the guidelines established for environmental matrices. Parameters such as linear concentration range, detection and quantification limits, accuracy, repeatability and matrix effect were evaluated. The criteria of the validation used in the work, previously published by the research group with HFO3E, will be applied [8].

*2.5. Behavior of the HFO4E and N$_2$ Gas Mixture in the Medium-Voltage Electrical Switchgear*

The medium-voltage switchgear used in this research are manufactured according to international standard IEC 62271-200:2011 "High-voltage switchgear and controlgear—Part 200: AC metal-enclosed switchgear and controlgear for rated voltages above 1 kV and up to and including 52 kV" [17] and IEC 62271-1:2017 "High-voltage switchgear and controlgear—Part 1: Common specifications for alternating current switchgear and controlgear" [18]. All of them have a hermetic tank made of AISI 304 stainless steel where the gas mixture is introduced.

Several prototype medium-voltage switchgear were filled with a gas mixture composed of HFO4E and N$_2$ at 1.4 bar, in the Ormazabal Corporate Technology experimentation unit (UDEX). The behavior of the gas mixture was evaluated in the following conditions:

2.5.1. Permanent Voltage of 30 kV

Three gas samples were taken from the cubicle of a medium-voltage switchgear filled with a gas mixture composed of 20% (*v/v*) of HFO4E and N$_2$: Prior to power-on, after one year, and after two years with permanent 30 kV AC. The samples were analyzed by GC/MS-TCD to assess the gas mixture stability and to determine the possible by-products formed. GC/TCD was used in order to detect compounds such as CO, CO$_2$, H$_2$ or CH$_4$, among others. The results obtained at the different times assayed were compared with those obtained with gas mixture standards.

2.5.2. Making Operations

Making operations according to IEC 622271-103 standard, at nominal current (100 × 630 A−24 kV) and at short-circuit current (5 × 20 kA−24 kV), were carried out in the High-Power Laboratory. A standard load-break switch interrupter is used in a 24 kV medium-voltage switchgear, filled with a gas mixture composed of 21.7% (*v/v*) HFO4E and N$_2$.

2.5.3. Dielectric Tests

Dielectric tests were carried out in one 24 kV medium-voltage switchgear filled with 21.7% (*v/v*) HFO4E and N$_2$ gas mixture, that is, at a power-frequency withstand voltage of 50 kV and at a lightning impulse withstand voltage of 125 kV.

2.5.4. Electrical Arc Tests

Analysis of HFO4E and N$_2$ gas mixtures in medium-voltage switchgear after an electrical failure producing an electrical arc was performed.

*2.6. Comparison of HFO4E and N$_2$ Gas Mixture Behavior versus SF6*

2.6.1. Physical Properties

The insulation gas used in medium-voltage switchgear must have a high dielectric strength but also must have a good capacity to evacuate the heat generated in the medium-voltage circuit, because of the current intensity passing through it. Therefore, dielectric strength, dynamic viscosity, thermal conductivity and specific heat, at 25 °C and 1 bar, of HFO4E and N$_2$ gas mixture were obtained using REFPROP software (https://www.nist.gov/srd/refprop (accessed on 1 September 2022)). The values were compared with those of N$_2$ and SF6 gas.

2.6.2. Breakdown Voltage

Breakdown voltage values of SF6, N$_2$ and different proportions of HFO4E (14–43%) and N$_2$ gas mixtures have been obtained experimentally with a BAUR DTA-100 system (https://www.baur.eu/en/home (accessed on 1 September 2022). This system consists of a hermetic recipient at a defined absolute pressure, with two electrodes separated by a distance of 8 mm, according to standard ASTM D2477. The voltage of one of the electrodes

is increased at 0.5 kV rms·s$^{-1}$ while the other is maintained at zero. This voltage increases up to a voltage level where the gas breakdown occurs.

### 2.7. OECD TG 403 Acute Toxicity Test

The initial toxicity of the gas mixture HFO4E and N$_2$ was obtained using the formula established in the international standard ISO 2098 for gas mixtures. The formula used and established by the ISO 2098 standard is represented by Equation (1):

$$LC50 = \frac{1}{\sum_i \frac{Ci}{LC50,i}} \tag{1}$$

where $Ci$ is the mole fraction of component $i$, and $LC50,i$ is the $LC50$ value of each component in the mixture. Since $LC50N_2 \geq 800,000$ ppmv and $LC50HFO4E \geq 25,400$ ppmv, the value of 108,047 ppmv will be the reference value of the initial mixture before toxicity assays in the case of the alternative mixture 21% HFOE and 79%N$_2$, $LC50$ mix, as reflected below:

$$LC50mix = \frac{1}{\frac{0.21}{25,400} + \frac{0.79}{800,000}} = 108,047 \text{ ppmv}$$

Making tests ($100 \times 630$ A $- 24$ kV and $5 \times 20$ kA $- 24$ kV) were carried out in a medium-voltage switchgear filled with an 21.7% ($v/v$) HFO4E and N$_2$ gas mixture with a positive result according to IEC 622271-103 standard. The acute inhalation toxicity of the gas mixture in the switchgear after these making tests was determined in the Toxicology Research Center (CERETOX), through the Experimental Toxicology and Ecotoxicology Unit (UTOX) of the Barcelona Science Park (PCB). According to method OECD TG 403, ten RjOrl:SWISS (CD-1®) mice, five females (3.9–40.3 g) and five males (39.8–54.9 g) of 12 weeks of age, were exposed during 4 h in a 10 L chamber with 30,000 ppmv of this gas mixture. The concentration of 30,000 ppmv was selected taking into account the acute toxicity result LC50 showed for SF6 after breaking tests at 630 A $- 24$ kV [19].

### 2.8. Influence of Temperature on HFO4E and N$_2$ Gas Mixture Behavior in the Switchgear

The IEC 62271-1:2011 standard of medium-voltage switchgear indicates that, in general, the temperature at any specific point must not exceed 115 °C, depending on the material and the dielectric gas used for the switchgear. Therefore, it is necessary to study if the temperature causes the generation of decomposition products on the gas mixture, as these could reduce the dielectric strength of the gas mixture or could be inflammable or even toxic.

The experiments were carried out using a system that was specifically created for this project. The designed experimental equipment has similar dimensions to an actual medium-voltage switchgear, but slightly narrower, 0.8 m $\times$ 0.5 m $\times$ 0.15 m. The experimental setup and its different components are shown in Figure 1. A heater was placed in the lower part of the gas chamber, which was connected to a temperature controller (Red Lion, York, PA, USA). Four temperature optic fiber measurement lines were located at four different heights inside the equipment and were connected to a temperature indicator (Weidmann Technologies, Dresden, Germany). The lines were named Top, Medium Top, Medium Bottom, and Bottom (T, MT, MB, and B) and were located at 0.71, 0.57, 0.35, and 0.13 m, respectively. The feeding system was composed of PTFE (polytetrafluoroethylene) tubes to feed and extract the gas mixtures in the chamber. A pressure indicator (Wika, Klingenberg, Germany) was used to control the composition of the gas mixture that was introduced.

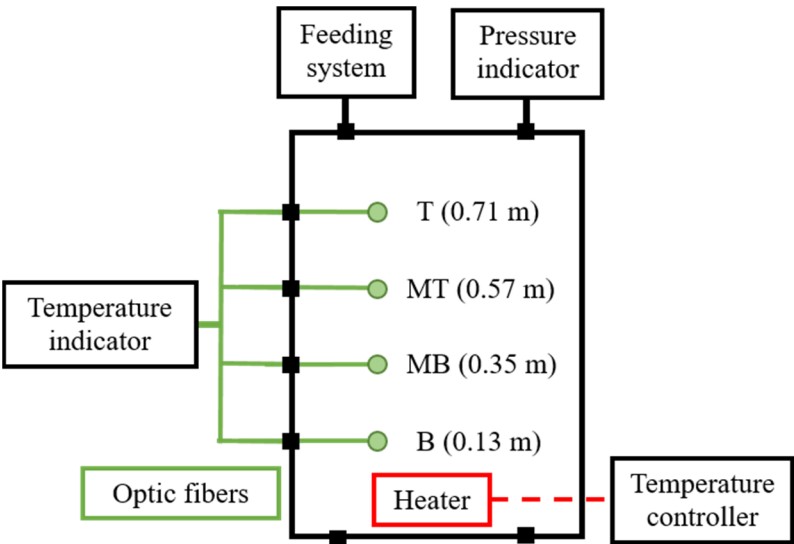

**Figure 1.** Scheme of the experimental equipment used for the monitoring of the temperature.

The gas chamber was filled up to atmospheric pressure, with 30% (*v/v*) of HFO4E and using $N_2$ as the carrier gas. Once the equipment was filled, the temperature controller was set to 120 °C. The heater temperature was maintained for 10 days.

## 3. Results and Discussion

### 3.1. Characterization of the HFO4E Standard

GC/MS and GC/TCD chromatograms for HFO4E standard, obtained under the experimental conditions previously optimized [8], are shown in Figure 2. Retention times of 18.5 and 17.8 min for GC/MS and GC/TCD, respectively and characteristic ions of 69, 95 and145 *m/z* were obtained.

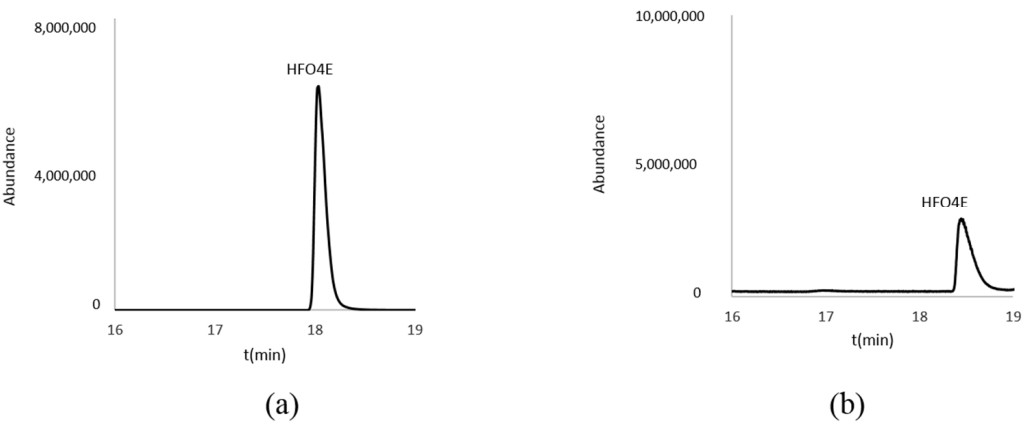

(a)                                                                                                (b)

**Figure 2.** (**a**) GC/MS and (**b**) GC/TCD chromatograms obtained for a gas mixture of 10% (*v/v*) HFO4E in $N_2$. Chromatographic conditions given in Section 2.3.

### 3.2. GC/MS-TCD Method Validation for HFO4E

The matrix effect was calculated by comparing the signals obtained from a standard of 20% (*v/v*), HFO4E in $N_2$ with a real sample from a low-voltage switchgear at the same concentration. The ratio between the two areas obtained was practically 1.0, so it was verified that there was no matrix effect and therefore this gas can be quantified by external calibration with HFO4E standards at different concentrations.

External calibration was linear up to 30% (*v/v*) HFO4E. The limit of detection (LOD) and quantification (LOQ) was calculated experimentally (Table 2) by introducing gaseous

mixtures of low concentrations. The chromatographic peak areas were interpolated in the calibration curve.

**Table 2.** LOD and LOQ for HFO4E obtained by GC/MS and GC/TCD.

| Analyte | LOD/LOQ (%)(*v/v*) | |
| --- | --- | --- |
| | **GC/TCD** | **GC/MS** |
| HFO4E | 1.00/1.50 | 0.15/0.25 |

The accuracy and repeatability of the method were obtained at the same day and at three different days, in terms of recovery (%R = value obtained*100/real value) and percentage of relative standard deviation (% RSD), respectively (Table 3). Two concentration levels were studied, 3% and 20% (*v/v*).

**Table 3.** Accuracy and repeatability values obtained for HFO4E by GC/MS and GC/TCD.

| | Concentration HFO4E (%) (*v/v*) | Accuracy Intraday/Interday (% R) | Repeatability Intraday/Interday (% RSD) |
| --- | --- | --- | --- |
| GC/MS | 3 | 99/98 | 5.0/6.1 |
| | 20 | 101/99 | 4.2/5.0 |
| GC/TCD | 3 | 98/98 | 4.1/5.5 |
| | 20 | 99/98 | 3.7/4.3 |

*3.3. Behavior of the HFO4E and $N_2$ Gas Mixture in the Medium-Voltage Electrical Switchgear*

3.3.1. Permanent Voltage of 30 kV

A complete overlap of the GC/MS and GC/TCD chromatograms was obtained for the HFO4E and $N_2$ gas mixture in the medium-voltage electrical switchgear, at different times assayed: 0, 1 and 2 years at 30 kV. Therefore, it can be deduced that the gas mixture is stable at least for two years. Chromatographic peaks corresponding to decomposition products were not detected.

3.3.2. Making Operations

Making operations ($100 \times 24$ kV − 630 A) and then short-circuit making operations ($5 \times 24$ kV − 20 kA) according to IEC 62271-103 were applied with positive results in a standard medium-voltage switchgear filled with the alternative gas mixture 21.7% (*v/v*) HFO4E and $N_2$. Two gas samples were extracted from the cubicle and analyzed by chromatography, the first one after $100 \times 24$ kV − 630 A making operations and the second one after the $5 \times 24$ kV − 20 kA making operations. In Figure 3, the chromatogram obtained after the application of the different making operations is shown ($100 \times 24$ kV − 630 A + $5 \times 24$ kV − 20 kA). Similar chromatograms were obtained for the first and second sample. $CF_4$, $C_2F_6$ and $C_2F_4$ products appeared in very low proportion. $C_2F_4$ is in a concentration of 0.0077% (*v/v*) and 0.0289% (*v/v*), respectively. From the results obtained, it can be concluded that the gas mixture is considered stable under these conditions.

3.3.3. Dielectric Tests

After dielectric tests in a 24 kV medium-voltage switchgear filled with 1.4 bars of 21.7% (*v/v*) HFO4E + $N_2$, the gas was barely deteriorated since the chromatograms of Figure 4 only showed the $C_2F_4$ product chromatographic peak. The proportion obtained of this product was very low, 0.0108%. (*v/v*). Therefore, the gas mixture can be considered stable under these conditions.

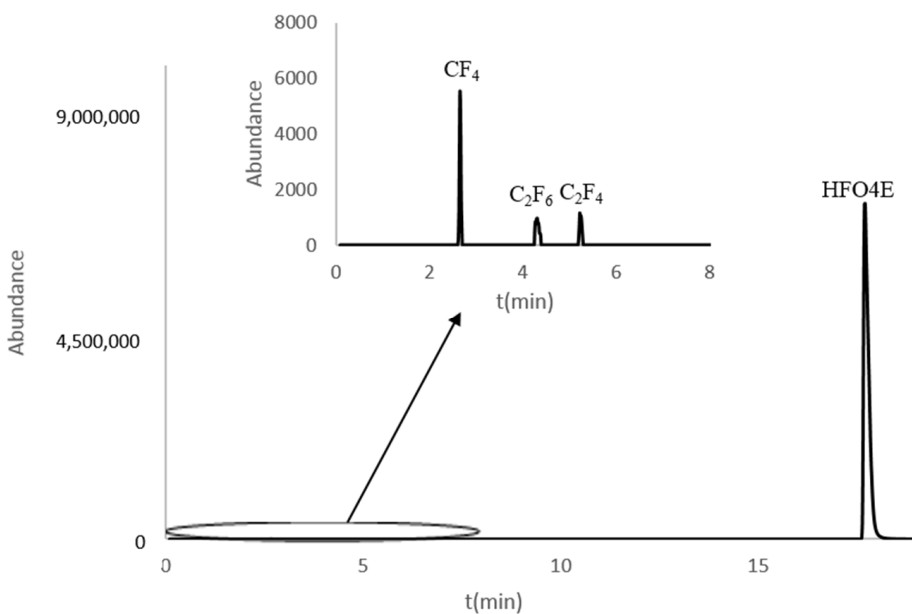

**Figure 3.** GC/MS chromatogram obtained for the gas mixture, 21.7% of HFO4E with $N_2$, after making operations ($100 \times 24$ kV $-$ 630 A $+ 5 \times 24$ kV $-$ 20 kA) in the medium-voltage switchgear.

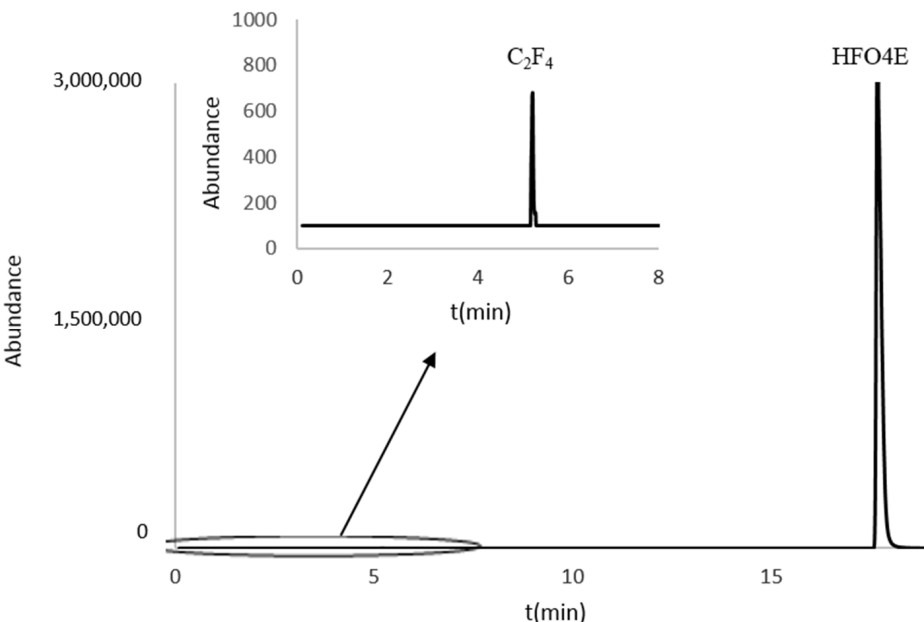

**Figure 4.** GC/MS chromatogram obtained for 21.7% (*v/v*) of HFO4E with $N_2$, after dielectric tests at 24 kV level.

### 3.3.4. Electrical Arc Tests

As can be seen in Figure 5, decomposition products of HFO4E were obtained after electrical arcs was produced, in a switchgear filled with HFO4E and $N_2$ gas mixture. At retention times of 6.0, 11.0 and 22.2 min appeared three chromatographic peaks corresponding to compounds not identified (N.I) by NIST14 library. The remaining decomposition products are collected in Table 4.

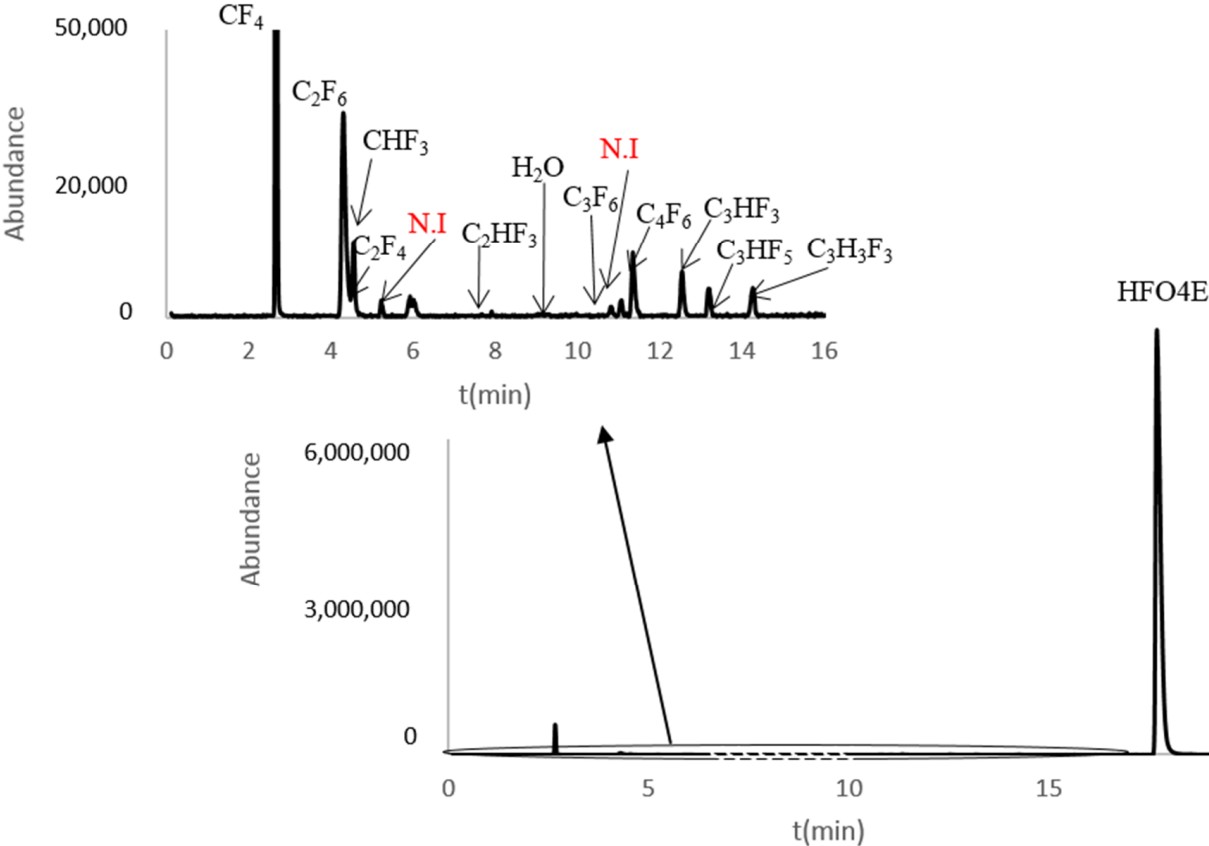

**Figure 5.** GC/MS chromatogram for a gas mixture of 21.7% (*v/v*) of HFO4E in $N_2$ after an electric arc of approximately 140 kJ.

No degradation products were found when the gas mixture samples were injected in the GC/TCD system. The percentage of chromatographic peak area of these degradation compounds with respect to HFO4E is negligible (Table 4) and, furthermore, the potential to cause cancer in humans of these compounds has not been assessed [20,21]. Their physicochemical characteristics and their toxicity are given in Table 4, with their retention times and chemical formula. Moreover, $C_3F_6$ and $C_2F_4$ were quantified with commercial standards, and 0.0140% (*v/v*) of $C_2F_4$ and 0.0090% (*v/v*) of $C_3F_6$ were detected.

$C_2F_6$, $C_2F_4$ and $CHF_3$ were also proposed as decomposition products in the theoretical study of Liu et al. [13]. For these authors, $C_2F_6$ is considered to be one of the most abundant decomposition products. $C_2F_4$ is discussed as a product of electrical failure and $CHF_3$ can form in the presence of micro-humidity. Apart from these three compounds, Liu et al. reported more than twelve molecules as decomposition products of HFO4E that were not detected in our studies.

### 3.4. Comparison of HFO4E and $N_2$ Gas Mixture Behavior versus SF6

#### 3.4.1. Physical Properties

In Table 5, the values of dielectric strength, dynamic viscosity, thermal conductivity and specific heat of HFO4E and $N_2$ gas mixture together with the corresponding to SF6 are collected.

As can be observed, the 21.7% (*v/v*) HFO4E and $N_2$ gas mixture has values of dielectric strength, dynamic viscosity, thermal conductivity and specific heat greater than SF6 values. These properties of the gas mixtures studied favor the heat evacuation of the switchgear. Dielectric strength of the gas mixture is lower than that of SF6. An improvement of the switchgear design has been necessary to achieve the standard dielectric assays.

**Table 4.** Retention times (tr) and physicochemical properties of decomposition products identified after the application to electric arcs of 20 and 140 kJ to the medium-voltage switchgear.

| Decomposition Products under Electric Arc | Chemical Formula | tr (min) | Chromatographic Area Relative to HFO4E (%) | Molecular Weight (g/mol) | GWP (100 Years) | Toxicity (LC50 4 h/rat, ppmv) | Flammability |
|---|---|---|---|---|---|---|---|
| $CF_4$ [4,9,21–25] PFC-14 Tetrafluoromethane | | 2.7 | 1.92 | 88.04 | 7390 | 89,500 * | Nonflammable |
| $C_2F_6$ [9,22,26] PFC-116 Hexafluoroethane | | 4.3 | 0.33 | 138.01 | 12,200 | 40,000 | Nonflammable |
| $CHF_3$ [9] HFC-23 Trifluoromethane | | 4.5 | 0.08 | 70.01 | 14,800 | N.A | Nonflammable |
| $C_2F_4$ [8] PFC-1114 Tetrafluoroethylene | | 5.2 | 0.01 | 100.01 | 4 [26] | 40,000 | Flammable |
| $C_2HF_3$ [9] HFO-1123 Trifluoroethylene | | 7.5 | 0.04 | 82.02 | 0.3 [24] | N.A | Flammable |
| $C_3F_6$ [3,9,22,25] Hexafluoropropylene | | 10.8 | 0.01 | 150 | <5 | 1672 | Nonflammable |
| $C_4F_6$ [9,23] Hexafluoro-1,3-butadiene | | 11.4 | 0.08 | 162.03 | <1 [24] | 1334 ** [25] | Flammable |
| $C_3HF_3$ [9] 3,3,3-trifluoroprop-1-yne | | 12.5 | 0.05 | 94.04 | 1.4 [26] | N.A | Flammable |
| $C_3HF_5$ [9] FC-1223zc 1,1,3,3,3-Pentafluoropropene | | 13.2 | 0.03 | 132.03 | <1 [26] | N.A | Flammable |
| $C_3H_3F_3$ [9] Trifluoropropene | | 14.2 | 0.04 | 96.05 | 0.12 [26] | 1750 ** | Flammable |

\* Value referred to 15 months, \*\* Value referred to 1 h, N.A. Not available.

**Table 5.** Values of dielectric strength, dynamic viscosity, thermal conductivity and specific heat of SF6, $N_2$ and 21.7% (*v/v*) HFO4E and $N_2$ gas mixture.

| Physicochemical Properties | Dielectric Strength (%SF6) | Dynamic Viscosity (cPoise) | Thermal Conductivity at 25 °C, 1 bar (W/mK) | Specific Heat at 25 °C, 1 bar (J/kg·K) |
|---|---|---|---|---|
| SF6 | 100 | 0.0145 | 0.0134 | 666 |
| $N_2$ | 45 | 0.0158 | 0.0260 | 1040 |
| 21.7% HFO4E and $N_2$ | 85 | 0.0154 | 0.0196 | 903 |

### 3.4.2. Breakdown Voltage

The breakdown voltage values of SF6 between 74–108% were obtained for the different proportions of HFO4E and $N_2$ gas mixtures (14–43%). In Figure 6, the comparison of breakdown voltages of SF6, $N_2$ and 21% HFO4E and $N_2$ gas mixture studied in this work is shown.

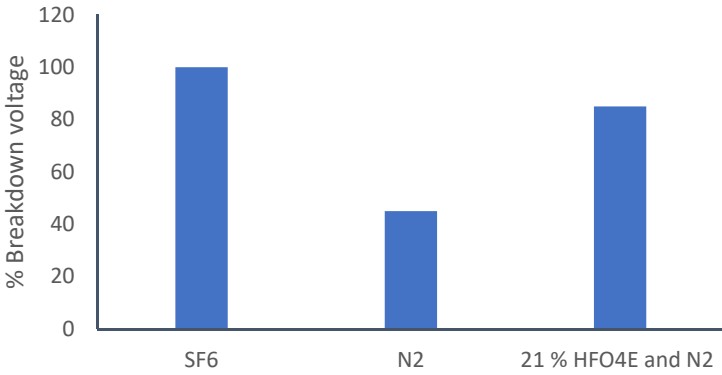

**Figure 6.** Breakdown voltage of the SF6, $N_2$ and 21% (*v/v*) HFO4E and $N_2$ gas mixture studied in this work.

### 3.5. OECD TG 403 Acute Toxicity Test

An acute toxicity inhalation test according to OECD TG 403 was performed with the gas mixture extracted from the medium-voltage switchgear after the making operations.

The result of the toxicological test carried out shows that the LC50 of the gas in the switchgear after the making tests carried out ($100 \times 24\ \text{kV} - 630\ \text{A}$ and $5 \times 24\ \text{kV} - 20\ \text{kA}$) is greater than 30,000 ppmv. Therefore, according to the classification of the Global Harmonization System, the gas after the making tests would not be classified as toxic. Monitoring of the mice for 14 days post-challenge also showed that the mice were still healthy and gaining weight.

### 3.6. Influence of Temperature on HFO4E and $N_2$ Gas Mixture Behavior in the Switchgear

The temperature profiles obtained for each measurement line are shown in Figure 7. It can be seen that it takes less than a day to achieve a nearly constant temperature at every height of the compartment. Additionally, a temperature gradient throughout the chamber can also be seen; the highest temperatures are at the bottom and they decrease when increasing the height, there being almost no difference in the temperature profile between the lines MT and T. The maximum obtained temperature is lower than 40 °C.

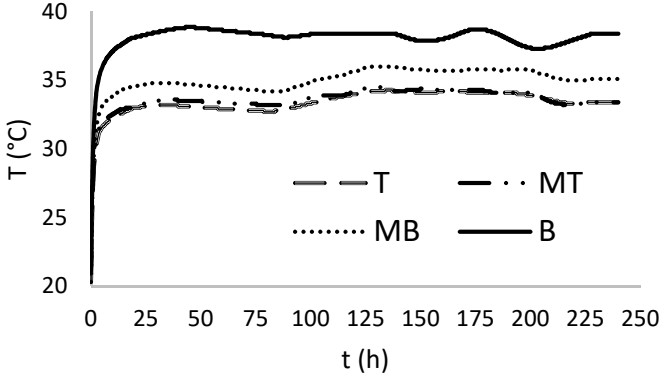

**Figure 7.** Temperature profiles of the measurement lines T (Top), MT (Medium Top), MB (Medium Bottom), and B (Bottom) for the whole experiment of a 30% (*v/v*) of HFO4E and $N_2$ gas mixture.

Two samples were taken on the 4th and 10th days of the experiment. The samples did not alter the composition of the gas mixture, since only 1 mL of gas was extracted each time—which is <0.002% of the total volume. The samples were analyzed by GC/MS and GC/TCD and no decomposition products of HFO4E were identified. Therefore, the mixture is considered to be stable at 120 °C for 10 days.

### 4. Conclusions

In this work, it has been shown that the mixture of HFO4E and $N_2$ is very stable under the following conditions: Under voltage at rated power frequency for long periods, high temperatures and making tests of rated currents and short-circuit currents. In the same way, no changes were observed in dielectric tests at 50 kV (power-frequency withstand voltage), at 125 kV (lightning impulse withstand voltage) and electric arcs of 140 kJ as a consequence of a dielectric failure. Furthermore, the acute toxicity assays with mice showed a LC50 greater than 30,000 ppmv for the gas mixture after the making tests. Therefore, neither the original gas mixture nor the resulting gas mixture after the tests would produce an effect on health and on the environment, if there were leaks in a medium-voltage switchgear with HFO4E and $N_2$.

As a result of this work, it can be concluded that the insulation properties of the medium-voltage switchgear were not affected by the different field conditions when the HFO4E and $N_2$ mixture is used. Thus, this gas mixture can be a good alternative to the use of SF6.

**Author Contributions:** Conceptualization, J.I. (Jesús Izcara) and J.I. (Josu Izaguirre); data curation, M.L.A., A.E. and J.I. (Jesús Izcara); formal analysis, M.L.A. and A.E.; funding acquisition, R.M.A., J.I.L. and J.I. (Josu Izaguirre); investigation, M.L.A. and A.E.; methodology, M.L.A. and J.I. (Jesús Izcara): project administration, R.M.A. and J.I.L.; resources, R.M.A. and J.I.L.; software, M.L.A. and A.E.; supervision, R.M.A. and J.I.L.; validation, M.L.A.; visualization, R.M.A.; writing—original draft, M.L.A., A.E. and J.I. (Jesús Izcara); writing—review and editing, R.M.A. All authors have read and agreed to the published version of the manuscript.

**Funding:** This work was supported by Ministry of Science and Innovation (project RTC2019-006844-3).

**Data Availability Statement:** Not applicable.

**Conflicts of Interest:** The authors declare no conflict of interest. The funders had no role in the design of the study; in the collection, analyses, or interpretation of data; in the writing of the manuscript; or in the decision to publish the results.

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
