# Peer review of "New Generation of SF6-Free Medium-Voltage Switchgear for the Electrical Network: Stability and Toxicity Studies of Trans-1,1,1,4,4,4-Hexafluorobut-2-ene with N2 Gas Mixture"

_processes, doi:10.3390/pr11010136_

Round 1

Reviewer 1 Report

The paper presents the study results of new generation of SF6-free medium voltage switchgear for the electrical network. Authors say, that binary gas mixture of N2 and trans hexafluorobut-2-ene (HFO4E), can be used as alternative to SF6 in high voltage devices. They tested its stability. Authors did not observed any changes in the gas mixture after AC voltage. They observed the same behavior under dielectric tests.

Dear authors, thank you very much for interesting paper about mixture, which eliminates SF6, which is perceived as greenhouse gas. I put some comments and questions.

Comments:

1. Introduction chapter is well organized and written. Please complete some information about SF6 as greenhouse gas. Authors just say that it is greenhouse gas, but they do not explain any properties of the gas, which prove that it is greenhouse gas.

2. Used references are correct and complete. I cannot see any auto citations.

3. Please use MDPI grammatical principles in the paper.

4. Material and Methods chapter is very complete and gives many important information, for example about used standards, which were applied to measure studied properties.

5. In main chapter 3, I would expect some comparison of investigated gas with SF6 or N2/SF6 mixture to make better conclusions. Please think about it.

6. Authors investigated dielectric properties. I did not see any characteristics, where dielectric properties of studied gas would be presented. Please complete.

7. General opinion. The paper topic is very interesting. Authors studied alternative gas in place of greenhouse SF6 gas. They say, that obtained results present dielectric properties, which I did not find in the paper. There are many important dielectric properties, such as breakdown voltage AC/DC, tan(delta), electrical resistivity, electrical permittivity. Also, thermal properties are very important in case of insulating gas, which will be used in high voltage devices, such as switchgear. I mean thermal conductivity, thermal capacity. Sorry, but I do not see mentioned properties in the paper. I suggest to complete the paper or to present obtained results in better methods.

Author Response

Thank you for your comments.

  1. Introduction chapter is well organized and written. Please complete some information about SF6 as greenhouse gas. Authors just say that it is greenhouse gas, but they do not explain any properties of the gas, which prove that it is greenhouse gas.

In Table 1 and in references 1 and 2 of the manuscript, GWP and information about SF6 greenhouse effect are included.

  1. Used references are correct and complete. I cannot see any auto citations.
  2. Please use MDPI grammatical principles in the paper.

The paper has been modified according to MDPI grammatical principles

  1. Material and Methods chapter is very complete and gives many important information, for example about used standards, which were applied to measure studied properties.
  2. In main chapter 3, I would expect some comparison of investigated gas with SF6 or N2/SF6 mixture to make better conclusions. Please think about it.

In the manuscript a new section entitled “Comparison of HFO4E and N2 Gas Mixture Behaviour versus SF6” has been included. This section has been divided in two parts. One dealing with physical properties and the other breakdown voltages of the SF6, N2 and HFO4E and N2 gas mixture. The properties of the gas mixture has been compared with SF6 and N2.

  1. Authors investigated dielectric properties. I did not see any characteristics, where dielectric properties of studied gas would be presented. Please complete.

Dielectric strength of natural origin gases (CO2, N2 or dry air) is significantly lower than that of SF6 at the same pressure. Therefore, to maintain size and footprint of actual medium voltage switchgear with SF6 gas without increasing gas filling pressure, new gas mixtures with higher dielectric strength than that of natural origin gases, must be considered as alternatives to SF6. One of these new gas mixtures is composed of hydrofluoroolefin trans-1,1,1,4,4,4-hexafluorobut-2-ene (HFO4E) and N2.

Breakdown voltage values of SF6, N2 and different “HFO4E +N2” gas mixtures have been obtained experimentally with a BAUR DTA-100 system (https://www.baur.eu/en/home) that consists of an hermetic recipient with two electrodes according to standard ASTM D2477 separated by a distance of 8 mm (Figure 1, PDF enclosed). During each test a gas or a gas mixture is introduced in the hermetic recipient with a defined absolute pressure and then the voltage of one of the electrodes is increased at 0.5 kVrms/s while the other is maintained at zero. This voltage increases up to a voltage level where the gas breakdown occurs.

The authors have not been considered the introduction of this figure in the manuscript.

The breakdown voltage values between 74-108 % were obtained for the different proportions of HFO4E and N2 gas mixtures (14-43%). In Figure 6 of the manuscript, the comparison of breakdown voltages of SF6, N2 and 21% HFO4E and N2 gas mixture studied in this work is showed.

  1. General opinion. The paper topic is very interesting. Authors studied alternative gas in place of greenhouse SF6 gas. They say, that obtained results present dielectric properties, which I did not find in the paper. There are many important dielectric properties, such as breakdown voltage AC/DC, tan (delta), electrical resistivity, electrical permittivity. Also, thermal properties are very important in case of insulating gas, which will be used in high voltage devices, such as switchgear. I mean thermal conductivity, thermal capacity. Sorry, but I do not see mentioned properties in the paper. I suggest to complete the paper or to present obtained results in better methods.

The values can be seen in the revised manuscript

Reviewer 2 Report

Please give at least a reference for this sentence if there is. “It is expected that at the end of 2023 a new F-Gas 30 Regulation will be approved in European Union to prohibit the commercialization of 31 switchgear with SF6 gas.”

In a table, dielectric (breakdown voltage, dielectric constant, partial discharge inception voltage, partial discharge extinction voltage) and thermal properties of the SF6 and your proposed alternatives should be compared.

Details about your test setup should be presented.

Potential of fire hazard should be investigated.

The authors should present dielectric characteristics of the nitrogen and other gases which introduced in this paper for this application. As examples, dielectric characteristics of nitrogen gas and liquid nitrogen can be accessed from below references:

HTS Transformer’s Partial Discharges Raised by Floating Particles and Nitrogen Bubbles. J Supercond Nov Magn 33, 3027–3034 (2020). https://doi.org/10.1007/s10948-020-05581-4.

The end part of cryogenic H. V. bushing insulation design in a 230/20 kV HTS transformer, Cryogenics, Volume 108, 2020, 103090, https://doi.org/10.1016/j.cryogenics.2020.103090.

Author Response

Thank you for your comments.

  1. Please give at least a reference for this sentence if there is. “It is expected that at the end of 2023 a new F-Gas Regulation will be approved in European Union to prohibit the commercialization of switchgear with SF6 gas.”

The following reference has been included in the manuscript:

Proposal for a Regulation of the European Parliament and of the Council on fluorinated greenhouse gases, amending Directive (EU) 2019/1937 and repealing Regulation (EU) No 517/2014    (05 April 2022):

https://eur-lex.europa.eu/resource.html?uri=cellar:ecf2b875-b59f-11ec-b6f4-01aa75ed71a1.0001.02/DOC_2&format=PDF    [see Annex IV (23)]

  1. In a table, dielectric (breakdown voltage, dielectric constant, partial discharge inception voltage, partial discharge extinction voltage) and thermal properties of the SF6 and your proposed alternatives should be compared.

In the manuscript one table of these properties has been included

  1. Details about your test setup should be presented.

All the tests were performed in gas insulated medium voltage switchgear. In this equipment the medium voltage circuit is located inside an hermetic stainless steel recipient that is filled to a pressure of 1.4 bar absolute with HFO4E+N2 gas mixtures as insulation gas instead of SF6 gas.

The medium voltage switchgear (filled in this case with 21.7 % HFO4E + N2 gas mixture) during the 100 x  24 kV-630A makings tests  according to IEC 622271-103 standard, in the High Power Laboratory of Ormazabal Corporate Technology in Amorebieta, Bizkaia (Figure 2, PDF enclosed).

These data, with the exception of the Figure 2, have been included in the manuscript.

  1. Potential of fire hazard should be investigated.

HFO4E is a non-flammable product (Table 1 of the manuscript) that has no low and high flammability limits. Therefore, gas mixtures of HFO4E+N2 are non-flammable.

  1. The authors should present dielectric characteristics of the nitrogen and other gases which introduced in this paper for this application.

In the manuscript one table of this properties has been included

As examples, dielectric characteristics of nitrogen gas and liquid nitrogen can be accessed from below references:

HTS Transformer’s Partial Discharges Raised by Floating Particles and Nitrogen Bubbles. J Supercond Nov Magn 33, 3027–3034 (2020). https://doi.org/10.1007/s10948-020-05581-4.

The end part of cryogenic H. V. bushing insulation design in a 230/20 kV HTS transformer, Cryogenics, Volume 108, 2020, 103090, https://doi.org/10.1016/j.cryogenics.2020.103090.

Thank you for the references you gave us, but the authors have considered including experimental values of the dielectric characteristics for N2, SF6 and the mixture studied collected in Table 5 of the manuscript. The values were calculated with a BAUR DTA-100 system (https://www.baur.eu/en/home).

Round 2

Reviewer 2 Report

new changes should be highlighted in the manuscript. details about the test setup should be added in the manuscript.

Author Response

In the current enclosed version of the manuscript all the changes done are highlighted. All the set up conditions are included. Please, specify if these changed are enough to replay you question or if you consider that more conditions are necessary.
